# The Effects of Processing and Preservation Technologies on Meat Quality: Sensory and Nutritional Aspects

**DOI:** 10.3390/foods9101416

**Published:** 2020-10-07

**Authors:** Inmaculada Gómez, Rasmi Janardhanan, Francisco C. Ibañez, María José Beriain

**Affiliations:** 1Departamento de Biotecnología y Ciencia de los Alimentos, Universidad de Burgos, 09001 Burgos, Spain; igbastida@ubu.es; 2Research Institute for Innovation & Sustainable Development in Food Chain, Universidad Pública de Navarra, Campus de Arrosadía, 31006 Pamplona, Spain; rasmi.janardhanan@unavarra.es (R.J.); pi@unavarra.es (F.C.I.)

**Keywords:** meat, processing, preservation, sensory quality, nutritional value

## Abstract

This review describes the effects of processing and preservation technologies on sensory and nutritional quality of meat products. Physical methods such as dry aging, dry curing, high pressure processing (HPP), conventional cooking, sous-vide cooking and 3D printing are discussed. Chemical and biochemical methods as fermentation, smoking, curing, marination, and reformulation are also reviewed. Their technical limitations, due to loss of sensory quality when nutritional value of these products is improved, are presented and discussed. There are several studies focused either on the nutritional or sensorial quality of the processed meat products, but more studies with an integration of the two aspects are necessary. Combination of different processing and preservation methods leads to better results of sensory quality; thus, further research in combinations of different techniques are necessary, such that the nutritional value of meat is not compromised.

## 1. Introduction

The changes produced in meat due to the application of different processing techniques, preservation methods, and technologies can be basically of two types: physical and chemical. Physical changes are modifications in the structure of the tissues that affect the sensory characteristics of the product such as volume, appearance, color, texture, aroma, and taste. Different effects in meat can be cited; reduced surface moisture due to dehydration, increased moisture and fat retention due to protein denaturation, and enhanced functional properties of proteins due to incorporated additives [1].

The chemical changes in meat are due to the molecular interactions that occur when thermal treatment is applied, food additives are added, or when storage is prolonged. When the chemical structures of the substances responsible for organoleptic characteristics or nutritional value are affected, for instance in the denaturation, hydrolysis, and gelation suffered by proteins due to the actions of boiling water and prolonged heating times [2], the consequences influence the consumer acceptance and affect balanced diet. Technologies which ensure food safety and meet the demands of the consumers without compromising the nutritional value of traditional meat products, are required.

Consumers demand preservative-free, minimally processed meat products with a longer shelf life. Nowadays the use of natural additives instead of synthetic additives is being widely accepted [3]. In addition to this, research on more ecofriendly packaging materials, which improves the shelf life of meat, is gaining momentum. The development of new meat products with improved nutritional profiles has increased over the last decade. For this purpose, there are two main strategies: obtaining healthier fresh meat and post-mortem processing of meat products [4]. These strategies could affect the quality of meat products and their nutritional value.

The meat processing method is usually selected mainly focusing on the technological, microbiological and healthy aspects of the product. However, when selecting a processing and/or preservation technology, not only the quality impact on the product should be considered; a comprehensive and global strategy considering the changes in sensory and nutritional features and consumer appeal is necessary. The objective of this review is to describe the effects of processing and preservation technologies on sensory and nutritional quality of meat products. The technical limitations, which arise due to the loss of sensory quality when nutritional value of these products improved, are presented and discussed. For the purposes of this review, only edible parts of terrestrial animals shall be considered meat.

## 2. Processing

### 2.1. Physical Methods

#### 2.1.1. Dry Aging

Dry aging is the process of ripening of meat at controlled conditions. The meat carcasses or primal cuts are hanged in a refrigerated chamber (0–4 °C) with the relative humidity maintained between 75 and 80% for 28–55 days. Until now, only bovine and porcine meats have been investigated. The process is comparably costly due to the need of quality meat cuts, shrinkage loss (6–15%), trim loss (3–4%), and the high risk of open-air contamination in meat. Open-air contamination can be reduced by packaging the meat in highly moisture permeable bags.

The effects of dry aging treatment on meat quality are summarized in Table 1. Dry aged meat has excellent flavor and palatability as a result of proteolysis, lipolysis, and concentration of flavor compounds due to water loss. Dry ageing imparts brown-roasted, beefy, buttery, nutty, roasted-nut, and sweet flavor in bovine meat [5,6]. In beef and pig, dry aged meat has an umami taste due to the high level of glutamate [7,8]. In a comparative study on dry aged and vacuum aged meat, it was observed that the umami and butter fried taste were more prominent in dry aged meat. Moreover, the consumer opinion on sensory aspects of dry aged meat was better in comparison to vacuum aged meat, the meat was found to be more tender and juicier. Dry ageing improves the tenderness and juiciness of bovine and porcine meat [6,7,9].

#### 2.1.2. Dry Curing

Dehydration is the process of reducing the moisture content in meat to improve its shelf life. Automated drying chambers with programmable logic controller and real time monitoring are nowadays widely used in the meat industry. In these chambers the air-flow rate, temperature, relative humidity, and flow distribution can be controlled relative to the size, shape, structure, and moisture content of the product [10]. The water holding capacity, the state of muscle proteins and its microscopic structure determines the rehydration property of the dehydrated meat. The muscle fiber diameter as well as the space between the groups of muscle fibers reduce during dehydration [11]. The rate of reduction in the moisture content during dehydration is high in precooked meat compared to raw meat. The heat damage during dehydration of meat is characterized by the burnt flavor, toughness, and grittiness. The conceptualization of the distribution of water in meat during dehydration can help optimize the process, which can be done by novel non-destructive techniques like hyperspectral imaging. Researchers have used the technique effectively in beef slices where the pixel wise images were taken at different time periods at six specific wavelengths [12]. Regarding the nutritional value of “dehydrated meat”, only two studies were carried out, and they were in the 1940s [13,14] (Table 1). They referred to dehydrated and packaged meat obtained with methods now in disuse. Most countries have their own traditional dried meat products, which have similar sensory features (Table 1). Kilishi is a traditional sun-dried meat product, which is spiced and roasted, with a shelf life of around 12 months. Biltong is another product where meat is salted and dried. Both products are commonly consumed in African countries. Carne do sol and charque are traditional salted dried Brazilian meat products. Tasajo, sou gan, pastirma, and cecina are salted and dried meat products traditionally prepared and consumed in regions like Cuba, China, East Mediterranean, and Mexico/Spain, respectively. Bresaola, jamón serrano, sucuk are traditional salted fermented and dried meat commonly consumed in Italy, Spain, and Turkey [15].

Dried meat products have a hardened texture and wrinkled appearance due to the volume reduction, and sometimes the meat has a hard crust on the surface. Aroma compounds are produced in the meat products as a result of lipid oxidation that imparts a characteristic flavor to the meat [16]. The dried meat has a brown color, the color changes from red to brown according to the temperature. The salt added during drying also adds to the darkening effect. Nitrate/nitrites can also be added to modify the color and flavor of the meat. In dry cured products, the characteristic flavor is due to the metabolites produced as a result of the action of enzymes on meat [16].

#### 2.1.3. High Pressure Processing

High pressure processing (HPP) is a non-thermal decontamination minimal processing technology, where the meat is subjected to a pressure range of 350–600 MPa for a few minutes to acquire improved microbiological safety and shelf life. Pressure is exerted isostatically and the volume of the product decreases with the increase in pressure. HPP can affect the sensory and nutritional characteristics of meat products (Table 1). Application of high pressure breaks the less strong ionic bonds and hydrogen bonds, which in turn denatures the protein via alteration of the quaternary structure of protein followed by tertiary structure at higher pressure ranges. The nutritive value of the meat is minimally affected by HPP [17]. The low molecular weight vitamins and flavor compounds stay intact since pressure does not affect covalent bonds [18]. High pressure treatment can potentially be an effective technology to improve the digestibility of meat products. This effect has been more pronounced in the muscles treated at 600 MPa [19,20].

A pressure higher than 200 MPa leads to the changes in meat protein with various effects of gelation, aggregation, and changes in texture due to the making and breaking of bonds. The effects also vary according to the range of pressure applied and the time of application of the pressure. Meat subjected to high pressure tends to conform to a gel consistency when the secondary and tertiary structure of protein breaks down keeping the primary structure intact. The characteristic structure of myoglobin changes with the application of pressure and it forms a new aggregated protein conformation with reduced solubility [21]. The elasticity of meat increases making it more tender [22]. HPP tends to modify the texture of the meat by tenderizing it, since high pressure fractures rod-like muscles in meat [23]. HPP induces unfolding of myofibrillar proteins, which subsequently exposes sulfhydryl and hydrophobic groups to the surface, unraveling helical structures and forming myosin oligomers through disulfide bond [24]. 

The meat treated with HPP at high pressure levels of 400 and 600 MPa were associated with browned, livery, and oxidized flavors [25], which will have an impact on the consumer and market behavior of the product. There is no immediate effect of HPP on the oxidative stress of meat [26]. Meat processed at higher pressures for longer period is tougher than the meat processed at a lesser pressure for lesser time [27]. The intrinsic properties of HPP processed meat differs according to the processing conditions applied and the type of raw material. HPP tends to induce aggregation, which improves the digestion of the meat [28]. Some studies did not observe any significant difference in the sensory properties of high pressure processed ready to eat (RTE) meat [29]. When smoked pork rounds were subjected to a pressure of 600 MPa for three minutes significant differences were observed in the cohesiveness and the odor of the meat, whereas the other textural and sensory properties were not affected by the pressure treatment [30].

The HPP treated samples of ham had paler color and softer texture compared to normal ham samples. Some studies concluded that HPP at 500 MPa combined with mild heat treatment at 53 °C was optimal for production of ham [31]. When goose meat was subjected to HPP at an optimal condition of 213 MPa for 15 min, it was observed that pressure range and the time of holding significantly affected the hardness, since the rod like muscles were fractured [23]. High pressure does not have much effect on cooking loss rate or water holding capacity. HPP at 450 MPa and 600 MPa did not significantly change the properties of seared beefsteaks in term of pH, water activity, moisture content, and expressible moisture [27]. However, enhanced water holding capacity was observed in rabbit muscles when subjected to HPP [26].

#### 2.1.4. Conventional Cooking

Cooking makes foods safe to consume and palatable. In order to guarantee food safety, food is cooked at higher temperatures for longer time; however, this practice decreases the nutritional and organoleptic quality of the foods; loss and oxidation of water soluble and thermolabile vitamins, loss of fats due to fusion, chemical browning reactions, etc. [32].

The effects of cooking temperatures on proteins are varied. At temperatures up to 100 °C, as occurs in water or microwave cooking, this denaturation translates into effects of interest, such as enzymatic inactivation of lipases, proteases, etc., improvement of digestibility or reduction of toxicity; between 100 and 140 °C, as in pressure cooking and baking, digestibility is reduced by forming intramolecular and intermolecular covalent bonds [33,34]. The same effects happen at temperatures above 140 °C, as in frying and roasting on the grill, where amino acid destruction occurs, such as cysteine or tryptophan, with isomerization to D-configuration and reduction of nutritional value. In lipids, heat treatment produces fusion, although being triglyceride mixtures it is difficult to establish its exact melting point; before reaching the liquid state, they go through a pasty state, then smoky (at a different temperature depending on the type of fat) and then decompose. Even intense heating can sometimes form toxic cyclic monomers, dimers and polymers, as is the case with acroleins. Carbohydrates are generally considered stable against cooking. However, solubilization losses of these compounds, which depend on factors such as time, size, etc., cannot be avoided [2].

With traditional cooking systems, there is a long waiting time between the preparation and the distribution of meals. Therefore, food must be placed in hot cabinets, ovens, water baths, etc., to avoid its cooling, where the food is dried and over-cooked. The result is a lukewarm meal, with a temperature below 65 °C in the center of the product, and therefore hygienically dangerous as these storage temperatures allow the growth of mesophilic and thermophilic microorganisms that will contaminate dishes during the waiting time for service and consumption. This fact can be dangerous in places of collective catering, such as hospitals, nursing homes, and school canteens, where the group to which the menu is directed has compromised immune system [35].

#### 2.1.5. New Techniques of Cooking: Low-Temperature Long-Time (LTLT) and Sous Vide Cooking

LTLT cooking has numerous advantages, of which the most sought-after characteristics are controlled doneness, improved tenderness and uniform eating quality. In LTLT cooking the product reaches a thermal equilibrium with the medium of heating, which contributes to these additional advantages of the product over traditional high temperature cooking. The underlying mechanism which provides more tender meat (Table 1), regardless of the age of animal, species, or type of muscle, at an optimum combination of temperature and time has not yet been completely elucidated. It might be, possibly, due to the interaction between proteolysis of myofibril structures and heat induced denaturation of proteins. The reduction of LTLT cooking temperature and holding time improve the juiciness of the meat, but at the same time in a constricted temperature range, higher cooking time imparts the desired aroma and flavor characteristics to the cooked meat [36]. The flavor intensity of the LTLT cooked meat is medium to low in comparison with the meat cooked at higher temperature [37,38]. The long cooking time weakens the forces holding the myofibrils together in aged meat leading to meat fragmentation upon shearing [39], and in meat with less amount of connective tissue the degree of tenderization is relatively high when cooked at 50–60 °C [40]. Prolonged heating time denatures the protein even if the temperature of cooking is lower than the actual temperature of denaturation [36].

Sous vide cooking (vacuum cooking) is new variant cooking technique used normally to produce high-quality dishes in the food service sector. Food is vacuum packed in a heat-stable plastic pouch, followed by incubation in a water bath at controlled conditions of time and low temperatures (53–81 °C) [41]. The cooking temperature is maintained lower with a higher cooking time. This technique maintains a uniform meat quality and improves the organoleptic property of the cooked meat. Sous vide cooked meat is more tender and redder than conventionally cooked meat. The duration and the temperature of the cooking comparably affect the physicochemical characteristics and palatability of meat [42].

Some studies have shown that there is significant effect of the cooking time and temperature over the texture of the meat. In sous vide cooked meat the increase of the cooking temperature and time result in increased shear force and toughening, respectively. However, the shear force is reduced when sous vide is combined with other treatments [43]. Water loss in the meat results in shrinkage of the muscular fibers both transversally and longitudinally, aggregation and gelling of sarcoplasmic proteins, shrinkage and solubilization of connective tissues, which leads to the formation of granular fibers. If the sous vide cooking is carried out at higher temperatures the cooking loss is maximum with a minimal reheating loss, due to the increased shrinkage caused by denaturation of the proteins [44]. In some studies of sous vide cooking, an increase of the opacity of meat surface was observed, which was due to the water loss. In sous vide cooked meat the reddish color of meat is replaced by a brownish red with a slight green color since the deoxymyoglobin and oxymyoglobin is denatured with an increase in the metmyoglobin and sulfmyoglobin as a result of longer cooking time [45]. The shelf life of sous vide cooked chicken tikka masala, a traditional Indian meat delicacy was comparatively high (40 days) with slight change in color. The higher shelf life was due to the spices and herbs in the product [46].

Sous vide processing renders meat juicier and more tender and at the same time, the technique improves the digestibility of meat, according to studies conducted on in vitro digestion [47]. However, in a study with young men, no differences were observed between the digestibility of sous vide cooked and fried meat in the pan [48]. Digestibility remains unknown in elderly adults. The volatile profile of sous vide cooked meat is better preserved with little accumulation of off flavor imparting compounds such as hexanal or 3-octanone usually found in traditionally cooked meat. A higher retention of vitamin B_3_ is another advantage of sous vide cooking since the cooking temperature is retained at a comparatively lower level [49].

Sous vide cooking can be carried out as low temperature long time or high temperature short time treatments. When the LTLT sous vide cooking method is used, collagen solubilizes, and a larger amount of gelatin is formed with less intense myofibrillar toughening [50]. The high temperature short time alternative can be considered as more economic and feasible method due to the higher safety and comparable quality attributes of the cooked meat, but with a lesser retention of vitamins and higher hardness [49].

Sous vide technique can be combined with other techniques, such as marination. For instance, Gómez et al. [51] reported the feasibility of using the combination of marination and sous vide cooking techniques to yield new RTE meat products with high protein content and without negative characteristics. In this way, the benefits of two different techniques are taken advantage without compromising the quality of the product.

#### 2.1.6. 3D Printing

3D printing is a novel pre-processing technology used in foods, where extrudable food is printed into specific shapes of uniform structure or layers using a 3D printer. There are several categories and ingredients of printable food (Figure 1). 3D printing enables automation, waste reduction, and personalization of foods. Meat should be processed into an extrudable form with added binders or texturizers, such as hydrocolloids or gellable proteins, so that meat can be 3D printed [52,53]. In addition to this, meat should be in a viscoelastic form so that it can be printed into the specific structure [53].

Few studies have studied the effects of 3D printing on sensory and nutritional characteristics of meat. Turkey puree with added binders and viscosity enhancer was successfully printed for sous vide cooking [54]. Researchers have conducted successful studies on the use of fibrous meat for 3D printing in medical field for elderly and patients who require ketogenic diet [55]. Moreover, beef paste prepared with guar gum binder and lard was 3D printed into multiple layers and sous vide cooked. The cooked samples maintained the structure with slight inward contraction in all the layers. It was observed that increasing the lard layers led to higher cooking loss, shrinkage, cohesiveness, lower fat retention, moisture retention, hardness, and chewiness, whereas increasing the infill density led to higher moisture retention with lower shrinkage and cohesiveness, resulting in higher hardness and chewiness [56].

The state of art in 3D printed meat is such that there are no scientific papers on nutritional and sensory properties, opening a huge opportunity for future research to focus on the same. Moreover post processing of 3D printed foods is required to render it edible, making possible to create a wide range of printed foods [53].

### 2.2. Chemical and Biochemical Methods

#### 2.2.1. Fermentation

Fermented meat products are mainly dry cured sausages, commonly eaten in many regions worldwide. In the Mediterranean regions, they are medium humidity meat products with considerable shelf life elaborated with spices such as paprika, garlic, and black pepper filled in casings and further cured or ripened so that the flavor would be enhanced. Prior nitrite treatment is considered as a mandatory pretreatment in most of the European countries [57].

The nitrogen compounds in the meat muscles are denatured enzymatically imparting meat the characteristic flavor. The enzymes like protease, aminopeptidases, and microbial enzymes breaks down the proteins in muscles, generating small peptides and amino acids, like alanine, leucine, valine, arginine, lysine, glutamic and aspartic acids, which imparts meat the characteristic flavor. In some cases, the stage of curing is assessed based on the concentration of these amino acids [58,59,60,61,62].

The effects of fermentation on the nutritional and sensory characteristics of meat are listed in Table 2. The flavor and quality of the finished product depends on the process duration. The color is determined by the amount of sarcoplasmic protein. The pH of the product decreases during the fermentation, leading to the gelation of the sarcoplasmic and myofibrillar proteins [63]. *Lactobacillus fermentum* was used as a substitute for nitrite in Harbin red Chinese style sausage, and it was observed that the characteristic pink color of cured meat was retained in the fermented meat [64].

The secondary oxidation products, formed as a part lipolysis and auto-oxidation in the lipids, develop specific aroma compounds like alcohols, aldehydes, ketones, esters, and lactones during the fermentation of the meat [63,65,66,67]. It has been found that meat proteins can produce bioactive peptides making it more susceptible to be used as functional ingredient too [68].

Presently in the food industry, starter and protective cultures are used, rather than relying on the natural microflora to ensure the sensory and microbial quality of the fermented meat products. During fermentation it has been found that various bacteriocins are produced, which inhibits the growth of other spoilage and pathogenic microorganism [69,70].

#### 2.2.2. Smoking

Smoking is an age-old preservation technique, where meat is subjected to smoke, which affects the sensory and nutritional characteristics of meat products (Table 2). There are positive effects, such as improvement of flavor, color and odor in lamb meat [71]. The effect of smoking on meat increases with the time of exposure [72]. Hot smoking, cold smoking, electrostatic smoking and use of condensates, smoke aromas or liquid smoke are different kinds of smoking treatments. Meat is smoked at 20–25 °C at a relative humidity of 70–80% and at 75–80 °C during cold and hot smoking, respectively. The electrically charged smoke particles which precipitate over the meat in electrostatic smoking reduces the time of processing [73]. In products where the protein denaturation which accompanies the smoking process is deemed undesirable, the smoke aromas or condensates are used [74,75].

Smoke process is an effective treatment against pathogenic microorganisms (*Staphylococcus aureus*, *Escherichia coli*, *Listeria monocytogenes*, *Salmonella* spp., etc.) [76], and reduces the lipid oxidation, which leads to undesirable flavors and oxidative rancidity [77]. In sausages smoking helps to reduce the greyish discoloration [78]. Smoking allows to incorporate different specie meats to obtain a high quality sensory sausages [79]. The sensory scores for smoked buffalo rumen meat products added with ginger extract were found to be within the acceptable limit during the storage period of 15 days [80]. Meat smoking enhances the sensory attributes but at the same time contaminates with carcinogenic residues such as polycyclic aromatic hydrocarbons or nitrosamines [72,81,82]. The residues are nowadays reduced by separating the smoke generating chamber and the smoking chamber, such that the residues precipitate in the generation chamber, exempting the meat of theses harmful residues [73].

Smoking reduces the water activity of meat, which effects the hardness of the product and the protein stability [73]. The digestible indispensable amino acids, which help in evaluating the protein quality in food, were calculated in smoked bacon and improved the content after smoking [83]. Curing combined with smoking has been found to increase the pH and improve the color, texture and odor of buffalo meat [84]. Smoke curing has the combined effect of both enzyme and heat, which leads to alterations in the fatty acid profile of pork and lamb meat [85].

#### 2.2.3. Curing and Salting

The main aim of adding salt into meat was to preserve the meat, but now the cured meats have a high demand due to their characteristic flavor and organoleptic properties. Nitrate and nitrite salts have been known to create the pinkish red color and characteristic flavor of meat and increase their shelf life. Nitrite salts inhibits the lipid oxidation that imparts a rancid flavor to meat [86] (Table 2). Recent studies have found a considerable amount of carcinogenic by-products formed as a result of adding nitrite to meat leading to the reduction in its use for curing [87] opening a window for the use of organically cured meats, where nitrate of natural origin from vegetable sources are used [88,89]. Smoking has been used to improve the preservative effect of meat during curing but nowadays due to the flavor imparted to meat, smoking has developed consumer appeal. 

Sodium chloride plays multifunctional role in preservation and processing of meat. It increases the shelf life of cured meat by reducing the water activity of meat, which in turn reduces the microbial load. Salt plays a critical function in determining the gelation, emulsifying and linkage properties in meat muscle proteins [90].

Scientific organizations lately propose the decrease of salt content in processed foods, which has led to a lot of studies on meat with salt replacements, such that the palatability and texture of cured meat are not compromised. Studies on Pamplona chorizo with low salt found that the product was acceptable [91]. The use of potassium chloride, flavor enhancers like carboxymethyl cellulose and carrageenan in combination with sodium citrate [92,93] or combination of sodium, potassium and magnesium salts [94], or undissolved salt crystals were used in various studies to reduce the sodium chloride content in meat with partial success [95], since low salt meat does not have the same palatability as the meat with normal sodium chloride content [96].

#### 2.2.4. Marination

Marination is a meat tenderization procedure, with the use chemical methods. This treatment increases the rate of natural proteolysis in meat, by means of a greater drop in the pH of the meat after slaughter, stimulating the enzymatic proteolytic activity during muscle maturation. Meat is treated with mixtures of different common organic acids like citric acid, acetic acid, and tartaric acid, from orange juice, apple cider vinegar, and agraz-verjus wine (unpublished results). It accelerates the maturation time of the meat by reducing the time necessary for its softening. The mechanism by which marinade influences meat tenderization appears to involve several factors including weakening of structures due to meat swelling, an increase in proteolysis caused by cathepsins, and an increase in the conversion of collagen to gelatin at a low pH during cooking [97].

When a piece of meat is directly immersed in an aqueous solution with various ingredients such as salt, organic acids, etc., the ingredients gradually penetrate by osmosis. The amount of salt and other ingredients from the peripheral parts of the final piece is superior to that of the central zones, not obtaining a homogeneous result [98]. That is why, at an industrial level, marination by immersion is replaced by marinade injection methods [99], which has several effects on the sensory properties of meat on or marinade injection technology in chicken and pork has been developed for years. The poultry industry has used water injection and polyphosphates for more than 20 years [100]; mainly with the aim of facilitating water retention during maturation and subsequent cooking, which leads to an increase in the juiciness of the meat and, with it, an increase in the perception of tenderness by the consumer. Researchers studied the acceptability and shelf life of fresh and precooked pork meat injected with salt, dextrose, citric acid, tripolyphosphate and sodium pyrophosphate, finding that, while citric acid and pyrophosphate lowers the pH of meat, tripolyphosphate increases it, causing a decrease in microbial growth and an improvement in sensory characteristics [101,102]. Researchers also used solutions of salts and phosphate in different cuts of beef, observing an improvement in juiciness and tenderness [103]. Injection at different pressures (345 and 200 kPa) implies differences in losses during cooking and in Warner-Bratzler shear force (WBSF) [104]. However, the softening and increase or enhancement of flavor by immersing the meat in a solution of different tenderizers or flavorings (marinated) has not been widely used in cattle; therefore, its real effectiveness is little known. Different authors have carried out brine injection tests containing, for example, sodium chloride, sodium tripolyphosphate and sodium lactate, finding an increase in juiciness over the control that had not been injected, but without finding significant differences depending on the proportion of injected brine with respect to the initial weight of the meat [105,106]. Other trials have focused on trying to alleviate some adverse effects observed as a result of brine injection, such as color loss or decreased shelf life [107,108]. 

Some authors have studied the effect of marinating beef with acidic aqueous solutions. For example, researchers investigated the softening of very fine beef cuts (40 × 35 × 5 mm), obtained from muscles with high connective tissue, from carcasses kept at least 48 h in refrigeration after the slaughter of the animal, and stored under vacuum at −20 °C until use [109]. With these cuts, they carried out immersion tests, for 20 h, in acid solutions prepared from acetic, citric or lactic acids, each one individually, or with mixtures of citrus juices (diluted orange and lemon). The results obtained led them to conclude that acid concentrations greater than 0.3 M were not recommended, as they caused excessive swelling of the meat, as well as its darkening and gelatinization.

Researchers also verified the effect of marinating pieces of neck meat, about 200 g, obtained from beef carcasses that had been maturing four days after the animal was slaughtered [110]. In this case, the marination solutions were prepared with acetic and lactic acids, in concentrations from 0.05 M to 0.25 M. Marination was carried out for two or nine days. Data on the shear resistance force, obtained with a Warner-Bratzler method, indicated that tenderness increased slightly at two days and slightly more at nine days, mainly due to proteolysis. In general, the higher acid concentration used for the marinade, the greater tenderness measured with the Warner-Bratzler method and the greater pH decrease in meat. However, tests carried out with panelists on the same meat indicated that marinating with a solution with an acid concentration greater than 0.15 M was not recommended, since the sensory panelists found the taste excessively acidic and rejected it. Meat with pH values lower than 5.0 was acceptable only up to a point.

#### 2.2.5. Reformulation

Basically two methods are possible to reformulate meat products: the elimination or reduction of components considered harmful to health (fat, saturated fatty acids, salt, nitrites, etc.) and the incorporation or increase of the content of substances with nutritional properties (dietary fiber, proteins of high nutritional value, polyunsaturated fatty acids (PUFA), monounsaturated fatty acids (MUFA), etc.). Table 3 summarizes different examples in which different reformulation alternatives and the consequences on the sensory and nutritional characteristics of foods have been studied.

##### Reduced Salt Content

The reduction of the salt content unfavorably affects some quality parameters of meat products such as sausages [111], bacon, cooked ham, and salami [112]. In order to minimize the negative effects related to its texture and the interaction of water and fat, binding agents such as phosphates, lactates, chlorides, alginate, and transglutaminase have been used [113], thus preparing various meat products using any of these alternatives (Table 1). The addition of phosphates in low-salt meat products improves the sensory and physico-chemical properties, since it increases the water and fat retention capacity, and reduces the salt content by up to 50% [114]. Gel-forming agents, such as calcium alginate or the enzyme transglutaminase, improves the binding properties and texture. The combination of different sodium, potassium, and magnesium salts have been found to produce meat products with acceptable sensory quality characteristics [114]. 

##### Fat Content Modification

The reduction of fat content is generally based on the use of leaner meat or the addition of water and resistant starches, non-starch polysaccharides, gums, or proteins [115]. For the development of low fat products, initial composition, desired final composition (percentage and type of fat) and the type of processing (cooking, curing, smoking, etc.) should be taken into account, since these factors affect the different quality attributes of the final product [114].

The main disadvantages of reducing the fat content in meat products are the loss of juiciness and obtaining a hard and rubbery texture. The solution is the use of different combinations of vegetable fats, proteins, and carbohydrates as fat substitutes, which mimic the mouthfeel and texture of the fat [116].

Reducing fat content in meat products does not reduce cholesterol content, and it has even been suggested that when fat is reduced and lean meat is increased, the cholesterol content of the meat product may increase [114]. Development of meat products with less cholesterol is based on replacing fat and lean meat with vegetable products that do not contain cholesterol, such as vegetable oils and plant proteins [117,118]. 

Meat products with a more suitable composition can be obtained by modifying the fatty acid profiles by using fats of vegetable and marine origin as partial substitutes for meat fats. In general, vegetable oils are rich in MUFA and PUFA and contain bioactive compounds. The fatty acid composition of the reformulated meat product will be affected by the type of oil used [119]. Various meat products have been elaborated using vegetable oils from olive, sunflower, cottonseed, corn, soybean, flaxseed, rapeseed, peanut, etc., and fish oils (Table 3). Although the replacement of animal fat by vegetable oils improves the lipid profile of the products [1,120,121], the percentage of fat that can be replaced without negative effects needs to be investigated.

Researchers studied the effect of using different proportions of olive and linseed oils for the total or partial substitution of animal fat on beef patties [121]. The best sensory results were obtained in beef patties when 50% of animal fat was replaced by 50% of a mixture of oils (25% olive oil and 75% linseed oil), which in turn gave rise to products with high content of *n*-6 and *n*-3 fatty acids of nutritional interest [121]. Likewise, consumers found no differences in the sensory parameters of these patties, with improved lipid profiles, with respect to the conventional patties [122].

##### Nitrite Content Reduction

In order to inhibit the formation of N-nitrosamines derived from nitrates added to meat products, sodium ascorbate, and erythorbate have been tested, but their effectiveness is limited due to the low solubility in adipose tissue. Studies have been conducted on the addition of fat-soluble derivatives of ascorbic acid, such as L-ascorbyl palmitate and long-chain acetals of ascorbic acid, the combination of α-tocopherol and ascorbate, and the use of lactic acid to inhibit the formation of N-nitrosamines [114].

Until now, a single compound to replace nitrites has been impossible to find, due to the multi-functional role they perform in meat products. Therefore, the solution is to combine several compounds that affect the color, flavor, antioxidant, and antimicrobial activity. Dyes such as erythrosine or the natural coloring pigments formed during external curing ("mononitrosil ferrohemochrome") can be used as alternative methods for maintaining the color of nitrite treated meat. The taste imparted as a result of the added nitrites is due to its antioxidant activity, which is why different antioxidants and chelating chemical agents can be used for its replacement. To replace the antimicrobial effect of nitrites, numerous compounds such as sorbic acid, potassium acid, sodium hypophosphite, fumaric acid esters, parabens, lactic acid producing bacteria, etc. can be used [114].

##### Incorporation of Protein and Dietary Fiber

Plant proteins are used in meat products to reduce costs, and to improve nutritional benefits [115]. Soybean and sunflower proteins, wheat and corn derivatives, cottonseed, and oatmeal flours have been used as fat substitutes in different meat products such as minced meats, hamburgers and sausages [114]. The functions of plant proteins in meat products are that they act as binders, improve the binding of water and fat and improve the water retention capacity [116]. Soybean protein has been used as a functional ingredient in different meat products like cooked minced meat and sausages [116].

Dietary fiber is incorporated into meat products due its health benefits and due to its ability to improve water and fat retention, increase emulsion stability, increase oxidative stability and modify texture [115,123].

Prebiotics such as inulin, a soluble dietary fiber, and products rich in dietary fiber have been used in the formulation of fresh, cooked, fermented, and crude-cured meat products [1,123,124,125]. The fiber-rich products that have been added come from many different sources, such as cereals, fruits, dried vegetables, roots, and tubers [115]. Adding dietary fiber improves nutritional properties by decreasing fat content, increasing fiber content, and maintaining sensory features [123].

#### 2.2.6. Enzymes

Enzyme applications include tenderizing meat, restructuring low-value pieces and trimmings of fresh meat for higher-quality products, and improving flavor and aroma. Table 4 summarizes different examples in which they have been used and the effects on the sensory characteristics of meat products.

##### Enzymes Used for Meat Tenderization

Natural proteolytic enzymes that improve meat tenderness can be from plant, bacterial, or fungal origin. The most widely used plant proteolytic enzymes to improve meat tenderness are papain, bromelain, and ficin [141]. The injection of papain into beef softens the meat, increasing tenderness, maintaining color, and organoleptic characteristics [142]. Injection of beef fillets with a bromelain solution increases tenderness tenderizes meat [143]. Treatment of mortadella with ficin softened the meat without modifying its organoleptic properties of the mortadella [144].

##### Enzymes Used for Meat Restructuring

Transglutaminase (TGase) improves meat texture characteristics, binding, and performance parameters. TGase can be used in meat emulsions to increase the binding of the solubilized proteins forming a stronger network, increasing the stability of the emulsion [145,146].

Moreover, TGase can bind meat of different shapes and sizes to obtain a uniform restructured meat product such as restructured cooked ham [147], low-salt chicken dumplings [148], chicken doner kebab [149], and restructured pork [150]. The production of restructured meat products with TGase is usually combined with the addition of proteins such as sodium caseinate [151], bisulfite-treated soybeans [152].

##### Enzymes Used to Produce Flavor and Aromas in Meat

The main enzymatic reactions that affect the flavor and aroma of meat products are proteolysis and lipolysis. These reactions can be carried out by endogenous proteases and lipases, enzymes of microbial origin naturally present in the product, or enzymes added during the manufacturing process [153].

The use of enzymes to improve flavor and aroma has been used mainly in cured meat products. During the maturation of cured meat products, proteolytic enzymes break down proteins and produce nitrogenous compounds and precursors of volatile compounds, which contribute to the development of flavor and aroma [154]. Lipases hydrolyze triacylglycerides into monoacylglycerides, diacylglycerides, and free fatty acids. Free fatty acids are oxidized to volatile aromatic compounds which contribute to the aroma of the final meat product [153].

The addition of proteases and lipases in the sausage and chorizo accelerates the proteolysis and lipolysis processes. However, researchers observed no improvement in flavor and aroma, and excessive softening occurred in the chorizo and in some sausages [155]. The addition of an extract of *Lactococcus lactis* and α-ketoglutarate in the salami produced an increase in the content of volatile compounds, and improved the sensory properties of the salami, increasing the flavor and aroma of the salami [156].

## 3. Preservation

### 3.1. Physical Methods

#### 3.1.1. Thermal Processing

Although cooking technique has been dealt previously in the corresponding processing section, this technique also has an objective in preservation, which is explained below. Pasteurization or sterilization of meat at high temperatures to attain a better shelf life, improved palatability, enhanced flavor is known as thermal processing. The time temperature combinations of thermal processing of meat is generally decided based on the required log reduction of the specific target microorganism, expected shelf life, and the physicochemical properties of meat [160]. The main target microorganism in thermally processed RTE food generally is *Clostridium botulinum*. The target organism in processed meat is *Listeria monocytogenes* since it grows at refrigerated conditions [161]. Cooking, sous vide cooking, canning, retort pouch processing, pasteurization are all different kinds of preservation techniques, which use high temperature to process and preserve meat.

Meat texture varies depending on the internal temperature applied during the processing and on the intrinsic characteristics of meat. The tenderness of meat increases at higher temperatures due to the denaturation of protein, solubilization of collagen, and formation of gelatin [162].

Thermal processing inactivates endogenous proteolytic enzymes and prevents development of off-flavors due to proteolysis. Heat sensitive vitamins are lost during prolonged heating at higher temperatures, but a comparable increase in the shelf life, flavor, and palatability is associated with heat treated foods. Oxidation of sulfhydryl group to disulfide group imparts the cooked flavor to meat. The color of meat changes when meat is subjected to high temperatures since myoglobin gets oxidized, which increases redness and reduces lightness of meat, relating it to the doneness of meat [163]. The consumer preference of the meat varies subjectively but the change in color is accepted as a preferred aspect of cooked meat.

Thermal processing reduces the oxidative stability of meat, which is detrimental to both nutritional and sensory quality of meat, but incorporation of antioxidants into meat has been proven to be a solution for the same [164,165]. Heat treatment has been found to concentrate the micronutrients like zinc, magnesium, iron, phosphorous in meat; however, some amount of these micronutrients are lost to thermal leaching [166]. 

A lot of studies have been carried out indicating the possibility of a modification of intramuscular fat in meat leading to an increase in the proportion of unsaturated fatty acids and therefore the nutritional value of meat [167,168,169].

RTE meat products tends to show acceptable sensory characteristics with a slightly reducing trend during storage; but these thermally processed foods in hermetic containers offers a shelf life of over 12 months with slight changes in sensory properties of meat. The sensory attributes like appearance, flavor, texture, juiciness, and overall acceptability of thermally processed traditional Indian meat curry, Rogan josh showed a significant declining trend during its shelf life of 12 months. Rogan josh is a meat product elaborate with thick gravy cooked with big chunks of meat added with spices and condiments, in the aforementioned study beef meat was used. The declining trend was due to the protein degradation and oxidation of the product [160]. Similar results were observed by researchers in RTE retort pouch processed Indian delicacies like Chettinad chicken [170].

#### 3.1.2. Packaging

Packaging options for meat and meat products are air permeable packaging, vacuum packaging, modified atmosphere packaging, active packaging, smart packaging, and edible coatings [171]. Table 5 summarizes different examples of packaging used in meat and meat products and their effects on sensory characteristics.

##### Vacuum Packing and Modified Atmosphere Packaging

Traditional packaging options for meat and meat products are air-permeable packaging, vacuum packaging, and modified atmosphere packaging. Modified atmosphere packaging improves meat color stability compared to vacuum storage [172], since the maintenance of the bright red color of the meat is possible; however, increased lipid oxidation may occur due to the oxygen content incorporated in the atmosphere. Although vacuum packaging inhibits lipid oxidation, which prevents the development of unpleasant smells and taste [173], the color of the meat becomes purple, which is not the bright red color that consumers associate with fresh meat. Thus, second skin vacuum packaging is considered better than conventional vacuum packaging because it is more visually appealing and has less weight loss [174]. Therefore, the combination of vacuum packaging and modified atmosphere methods can be an efficient way to reduce the negative quality changes that occur when using these systems separately [174].

##### Active Packaging

Active antioxidant packaging controls the oxygen levels of the packages. There are two active antioxidant packaging systems: separate antioxidant devices and packaging materials with built-in antioxidants [171]. Stand-alone antioxidant devices are sachets, pads, or labels that contain oxygen scavengers. The active agent is incorporated into the walls of the packaging material, so that undesirable compounds are absorbed from the headspace or the antioxidant compounds are released into food [171]. An advantage of incorporating antioxidants in packaging materials, compared to direct addition to food, is controlled release of the active compound.

Antimicrobial agents that have been used in active antimicrobial packaging include plant and spice extracts, essential oils, peptides, organic acids, antibiotics, bacteriocins, and silver ions [163,175,176]. Essential oils have been widely studied as natural antimicrobial agents for meat and meat products, but they produce intense aromas and flavors that affect the sensory quality of meat [177]. Therefore, novel technologies such as the encapsulation of essential oils in nano-emulsions, the incorporation of essential oils in nanorcils and the use of essential oils combined with other antimicrobial methods or agents have been developed. For instance, researchers designed an antimicrobial active packaging system for RTE meat products, consisting of a film of chitosan with thyme essential oil, which improved the color of the meat and prevented the appearance of unpleasant odors [178]. Likewise, the use of chitosan coatings did not influence the sensory characteristics of chicken meatballs and chicken skewers [179] or fresh pork sausages [180].

##### Intelligent Packaging

The most widely used smart devices in packaging are barcodes, radio frequency identification labels, time–temperature indicators, gas indicators, freshness indicators, and pathogen indicators. Oxygen indicators are the most widely used gas indicator in meat products, since oxygen can cause oxidative rancidity, color changes and development of pathogenic and altering microorganisms [171]. The freshness indicators are based on the detection of freshness indicator metabolites, whose presence causes a color change [181].

This type of smart packaging does not prevent the degradation of the products [182], it only helps to maintain the physical and sensory characteristics, and inhibits degradation resulting from oxidation reactions or product contamination. Therefore, the combination of smart packaging with other technologies is necessary. Researchers studied the combination of smart and active packaging technologies and stated that this combination can be used to assist in the modification of conventional packaging systems in order to enhance product quality and safety [183]. However, further studies would be necessary to assess the impact of the combination of smart packaging technology with other packaging technologies on the organoleptic quality of meat and meat products.

##### Edible Films and Coatings 

The films are made with biopolymers that are based on hydrocolloids such as polysaccharides, proteins of animal origin and proteins of plant origin. Films formed are impervious to moisture and gases but have worse mechanical and functional properties than plastic films [183]. Chitosan is a biopolymer with antioxidant and antimicrobial properties and can be used as a matrix to develop edible films [184]. Edible films and coatings for meat packaging can be combined with the incorporation of active components with antioxidant and antimicrobial properties. The active components that can be incorporated in edible films are natural extracts, essential oils, natural polymers, protein hydrolysates, enzymes, and nanocomponents [185].

#### 3.1.3. High Pressure Processing

HPP is also applied for preservation purposes, although HPP has been dealt previously in the corresponding processing section, this section is focused on the conservation approach. Response of vegetative pathogenic and spoilage microorganisms to HPP depends on process parameters such as pressure, temperature, processing time, and on product parameters such as pH, water activity, salt content, and the presence of other antimicrobials. Inactivation of more than four log units of common vegetative pathogenic and spoilage microorganisms can be attained by HPP at 400–600 MPa with short processing times of 3–7 min at room temperature [22].

Ham samples treated by HPP showed increased hardness and syneresis during storage [28]. Ham samples HPP treated at 600 MPa for 5 min showed 2 *log* and 3 *log* reductions of *L. monocytogenes* on the surface and interior respectively. Treatment at 600 MPa for 5 min in completely dry-cured ham reached the food safety objective for *L. monocytogenes,* without significantly affecting the physicochemical characteristics of dry-cured ham [197].

### 3.2. Chemical and Biochemical Methods

#### 3.2.1. Food Additives

The main additives used in the production of meat derivatives are antioxidants, binders, antimicrobials, curing agents, and curing accelerators.

Synthetic antioxidants approved for use in meat products are butylhydroxyanisole (BHA), butylhydroxytoluene (BHT), propyl gallate, terbutylhydroquinone (TBHQ), and tocopherols. These antioxidants delay or inhibit the oxidation of meat and meat products, and therefore avoid the appearance of unpleasant odors and flavors.

Binder additives are added to the meat to maintain a uniform dispersion of fat throughout the product and to prevent water loss during the different stages of processing, heating, storage, and cooling. The binder additives used in meat and meat products are phosphates, starches, xanthan gum, guar gum, sodium alginate, carrageenan, carboxymethylcellulose, etc. It is worth highlighting the functions of phosphates in meat products, which are to increase the water retention capacity and increase the stabilization of the emulsion. Phosphates also have other functions such as stabilizing color, inhibiting lipid oxidation, and promoting protein dispersion [197]. 

The synthetic additives used as antimicrobials in meat products are organic acids such as acetic, lactic, propionic, sorbic, benzoic, and citric acids, and sulfites. Sulfites have antimicrobial activity against decomposing microorganisms; however, sulfites cause health problems such as allergic reactions in sensitive people. Organic acids have activity against a wide variety of pathogenic and disrupting microorganisms [198]. Sorbic acid is used in meat products for its inhibitory activity against yeasts and molds. However, it does not affect lactic acid bacteria, which makes it useful as a preservative in fermented meat products [199].

Nitrates and nitrites are the most widely used curing agent in meat products. Nitrites provide the red color and flavor of cured meat, and have antioxidant and antimicrobial properties. The reduction of nitrites and nitrates to nitric oxide is important for the color of cured meat. Nitrosyl myoglobin, which is the dominant pigment in cured meat products, is formed from the interaction of nitric oxide with the *heme* group of myoglobin [200]. Curing accelerators such as sodium ascorbate, sodium erythorbate, ascorbic acid, or erythorbic acid are added to meat to speed up the curing process, as they reduce nitrites to nitric oxide. Nitrites react with amines and amino acids leading to the formation of N-nitrosamines, which are chemical agents with potentially carcinogenic, mutagenic and teratogenic activities. For this reason, alternatives are being sought to reduce or eliminate the addition of nitrites in meat products and reduce health risks [201].

#### 3.2.2. Natural Antioxidant Ingredients

Natural antioxidant ingredients can be used as food additives in meat and meat products for their technological properties. Table 6 summarizes the effect of the addition of natural antioxidants in the formulation of meat and meat products on their sensory characteristics. The antimicrobial and antioxidant activities of some plant extracts and/or their essential oils are mainly due to the presence of some major bioactive compounds, including phenolic acids, terpenes, aldehydes, and flavonoids [202]. It should be highlighted that natural antioxidants can be incorporated into packaging systems, which has been previously explained in the corresponding section.

##### Essential Oils and Spices

In some cases, the addition of essential oils or spices can have negative effects on meat. For example, the addition of essential oils of oregano and thyme in lamb meat in concentrations greater than 1%, produces a strong odor and unpleasant taste in the product [203]. Adding different extracts like clove or cinnamon to raw chicken can increase *L**, *a**, and *b** values during storage [204]. Cinnamon also increases the redness of meatballs, although it does not affect the other sensory characteristics [205]. The addition of turmeric powder to rabbit patties changed the color of the meat, due to the yellow color of turmeric [206]. Therefore, studies of the concentrations of the added oils and spices, to avoid these negative effects on the sensory characteristics, are required. Adding clove extract to cooked and refrigerated beef patties increased the patties’ red color and maintained the sensory characteristics for up to 10 days of storage [207]. Addition of cumin, clove, and cardamom in rista, a traditional Indian sheep meat product cooked with spices, improved the shelf life of the meat product to 25 days with a high overall acceptability score [208].

##### Plant Extracts

The antioxidant activity of rosemary extract has been evaluated in pork burgers, and it was observed that it did not develop any adverse effects on sensory characteristics or general acceptability of the product [209]. The addition of rosemary (0.25% *v*/*w*) did not negatively influence the taste of turkey meat [210]. The incorporation of oregano extract in sheep burgers did not affect the sensory properties of the burgers [211]. No comparable difference in aroma, flavor, and overall acceptability was observed in low-salt sausages when garlic derivatives were added to it [212]. 

The addition of green tea extracts to low sulfite beef patties delayed the appearance of rancid flavors, decreased the loss of red color, and did not modify the odor, taste, and texture of the patties [213]. The addition of 250 mg/kg of grape seed extract did not affect sensory characteristics or instrumental color in beef enriched with *n*-3 and CLA [139,214]. However, adding grape seed extracts and green tea could darken pork meatballs [215]. 

Sensory quality was not negatively altered when blueberries were added to pork burgers and cooked pork ham [216], or when raspberry pomace extracts were added to beef burgers [217], or extract of pomegranate peel and pomegranate juice in cooked chicken patties [3].

The wine pomace, also called grape pomace, a by-product from winery rich in antioxidants [218] is used in the preparation of meat products [219]. For instance, red wine pomace may be an alternative to sulfites as a meat additive for protection of beef patties against protein oxidation [220]. However, the presence of anthocyanins in red grapes has the disadvantage of darkening the product, which could modulate consumer opinion, what must be studied for every type of product. For example, the excess color of seasoned nuggets did not adversely affect the evaluation of other sensory characteristics such as juiciness, crispness, oiliness, saltiness, and chicken flavor [221].

The byproducts of citrus fruit juice processing can be considered as potential ingredients in meat products because of their ability to reduce residual nitrite levels, thus avoiding the possible formation of nitrosamines and nitrosamides. The addition of citrus fiber washing water did not affect the color or texture properties of Bologna sausage, and its combination with rosemary essential oil led to the best sensory quality [222].

## 4. Conclusions

Numerous techniques have been developed to obtain healthier meat and meat products. However, the modification of the sensory quality should be considered when processing and preservation technologies are applied. Combinations of different technologies are necessary to achieve the best sensory quality in products with improved nutritional profile. In the development of new meat products, a comprehensive approach, including evaluation of sensory characteristics and nutritional value is necessary. Consumers are currently looking for minimally processed products, which are environmentally sustainable. Thus, this review describes the different alternatives, with their advantages and disadvantages, highlighting techniques that improve the sensory characteristics and give rise to products with improved nutritional profile and consumer appeal. Among the processing techniques, physical treatments such as dry aging and HHP stand out as they allow intensifying the flavor and increasing the tenderness of the meat products. Another interesting physical treatment is sous vide cooking, as it is an appropriate technique to preserve nutrients and maintain a high organoleptic quality. Preservation techniques of natural origin stand out as they prolong the shelf life of meat products without negatively affecting the sensory features. The combination of these techniques will make it possible to expand the offer of meat products elaborated with original raw materials, maintaining or even improving their nutritional and sensory characteristics for long periods of time.

## Figures and Tables

**Figure 1 foods-09-01416-f001:**
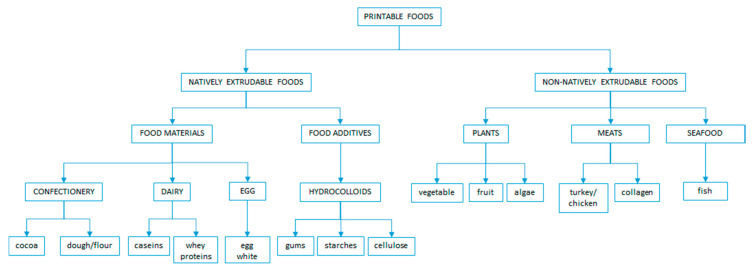
Printable food classification, categories and ingredients.

**Table 1 foods-09-01416-t001:** Effects of some physical treatments on the sensory and nutritional characteristics of meat products.

Treatment	Meat Product	Effects
Dry aging	Beef meat [5,6,8,9] Porcine meat [7]	More flavor, tenderness and juiciness in beef. Umami taste in beef and porcine meat. Nutritional changes not investigated.
Dry curing	Pork, beef, mutton [13]	Increased storage temperature. slightly decreased the digestibility of dried pork protein.
	Meat products [14]	Protein quality is not significantly reduced during dehydration.
	Meat products from different animals [15,16]	Hardened texture, wrinkled appearance, characteristic flavor, brown color and darkening.
High pressure processing	Beef, pig, chicken meat [17]	Unchanged nutritional value.
	Different meat products [18]	Low molecular weight vitamins and flavor compounds stay intact.
	Beef [19] and rabbit [20] muscle	Enhanced digestibility.
	Meat products [22]	Improved tenderness, changes to the color quality. depending on the content of myoglobin.
	Goose breast [23]	Improved tenderness.
	Lamb meat cuts [25]	Browned, livery and oxidized flavors.
	Ham [29]	Improved digestibility.
	Ready to eat (RTE) meat products [30]	No changes in sensory properties.
	Pig ham [32]	Paler color and softer texture.
Low-temperature long-time (LTLT) and sous vide cooking	Meat [36]	Increased tenderness and better appearance.
	Lamb [37] and pork [38] meats	Increased flavor.
	Beef [39,40]	Increased tenderness.
	Chicken meat [42]	Increased tenderness and color.
	Beef [45]	Brownish red with a slight green color.
	Pork [47]	Juicier and more tender meat, and improved digestibility.
	RTE marinated beef [51]	No effects on sensory characteristics.
3D printing	Turkey and beef meats [54,56]	Novel appearance and texture. Nutritional changes not investigated.

**Table 2 foods-09-01416-t002:** Effects of fermentation, smoking, curing and salting, and marination on the sensory and nutritional characteristics of meat products.

Treatment	Meat Product	Effects
Fermentation	Dry-cured meat products, traditional Jinhua ham, Parma ham, dry-cured Iberian ham [64,66,68]	Specific aroma compounds such as alcohols, aldehydes, ketones, esters and lactones.
	Harbin red Chinese style sausage [64]	The use of *Lactobacillus fermentum* led to characteristic pink color of cured meat.
	Fermented meat products [68]	Production of bioactive peptides.
Smoking	Lamb meat [71] Sausage [78] Sausages from poultry, pork and beef meat [79] Smoked pork bacon [83]	Enhanced flavor, color and odor. Reduction of the greyish discoloration. Enhanced sensory attributes. Improvement of digestibility of indispensable amino acids.
	Buffalo meat [80]	Combination of smoking with curing improved color, texture and odor.
Curing and salting	Bovine muscle [90]	Improved texture properties.
Marination	Broiler chicken [100] Fresh and precooked pork meat [101,102]	Increase of the juiciness and tenderness.
	Beef [103]	Tripolyphosphate in brine improved sensory characteristics.
	Beef [107,108]	Brines of salts and phosphate improved juiciness and tenderness. Color loss.
	Beef [109]	Acid concentrations greater than 0.3 M were not recommended, as they caused great swelling and darkening.
	Beef [110]	The higher acid concentration used for the brine, the greater tenderness. Solution with an acid concentration greater than 0.15 M lead to too acidic beef and rejection by panelists.

**Table 3 foods-09-01416-t003:** Effects of the reformulation of meat products on their sensory and nutritional characteristics.

Compound	Reformulation Objective	Treatment	Meat Product	Effects
Salt	Reduction	Lowering from 2.8% to 0.5%	Hotdog sausages, bacon, ham and salami [112] Pork sausages [126] Chicken breasts [127]: Reconstructed ham [128]	Paler, softer, and less juicy products per low of 1.3–1.7% NaCl. Difficulty reducing the dietary salt intake (<1.4%) without affecting acceptance.
	Partial substitution	Use of spice mixes, KCl or other salts	Cooked ham [92] Bovine and chicken meat [129] Fermented sausages [111,130] Frankfurt sausages [131]	Sensory quality and general acceptability were not modified if replacement ranged 30–35%. Reduced acceptance of aroma, flavor, juiciness and overall quality if NaCl was lower than 1.3%.
Fat	Reduction	Addiction of vegetable oils	Pork sausages [117] Frankfurt sausage [126] Pork sausages [132] Beef and pork sausages [133] Beef patty [134]	Darker, harder, less juicy and less flavor intensity. Better nutritional value (reduction in fat and cholesterol and increase in polyunsaturated fatty acids (PUFA) or monounsaturated fatty acids (MUFA).
	Substitution	Replacing by vegetable or fish oils, soybean proteins, carbohydrates, and synthetic compounds	Sausages, cooked minced meat [116] Veal sausages [118] Bologne sausages [120] Beef patty [121] Spanish salami [135] Sausage [1]	Decrease of meat aroma and flavor intensity. Better nutritional value (reduction in fat and cholesterol and increase in PUFA or MUFA).
	Enhanced nutritional value	Raw material with a high level of mono and polyunsaturated fatty acids from pigs fed with different diets	Dry fermented sausage salchichon [136]	The color was slightly affected. Improved nutritional value.
		Grass-fed or flaxseed-containing concentrates	Beef [137,138]	Improved fatty acid profile by increasing content in conjugated linoleic acid (CLA), eicosapentaenoic acid (EPA) and docosahexaenoic acid DHA.
		Feeding with linseed seeds and CLA	Beef patty [139]	No significant change in color and odor of hamburgers enriched in *n*-3 and CLA. Enhanced lipid profile.
Dietary fiber	Addition of dietary fiber	Addition of dietary fiber (inulin, rice fiber, citrus fiber, etc.)	Sausages [1] Meat products [123] Roast beef [124] Bologne sausages [125,140]	Texture properties decreased (harder and less chewy structures). The 6% inulin concentration provided the best sensory characteristics. Maintained the sensory properties and acceptability. Nutritional value in PUFA improved, the fat content decreased, and the fiber content increased.

**Table 4 foods-09-01416-t004:** Effects of some enzymatic treatments on the sensory characteristics of meat products.

Objective	Treatment	Meat Product	Effects
Tenderization	Addition of papain, bromelain, ficin,	Beef meat [142] Mortadella [144] Turkey, hen and rooster thighs [157] Beef cubes [158]	Increase of tenderness. Without changes in organoleptic properties.
	Blade tenderization, bromelain or salt/phosphate injection	Muscles from beef rounds [143]	Injection with a salt and phosphate solution resulted in the lowest Warner-Bratzler shear force (WBSF) values. WBSF values for blade tenderization and enzymatic tenderization were comparable.
Restructuring	Addition of transglutaminase (TGase)	Restructured cooked ham [147] Pork gels [159] Low-salt chicken dumplings [148] Chicken sausages [145] Doner kebab of chicken [149] Sausages and ham [146] Restructured pork [150]	No effect on color. Formation of network structures, improving the textural properties: increase of springiness, firmness, decrease in adhesiveness. Increased juiciness, tenderness and overall acceptability. Increase of firmness of meat gels.
	Bisulfite, soybean protein and TGase	Pork sticks [152]	Improvement of tensile strength and cooking performance.
	Sea spaghetti seaweed (3% dry matter) combined with NaCl reduction and a (TGase/caseinate) system	Restructured poultry steaks [151]	Increase in Kramer shear force. Products were acceptable.
Production of flavor and aromas	Addition of palatase M and protease P	Spanish dry fermented sausage (Pamplona chorizo) [155]	Without changes in the sensory quality except a slight softening.
	Intracellular cell free extract (*L lactis* NCDO 763,) and α-ketoglutarate	Dry fermented sausages [156]	Improvement of odor and flavor when *L. lactis* and α-ketoglutarate were combined.

**Table 5 foods-09-01416-t005:** Effects of different types of packaging on the sensory characteristics of meat products.

Treatment	Meat Product	Effects
Air-permeable packaging, vacuum packaging and modified atmosphere packaging	High pH and normal pH beef [172] Lamb slices [173] Beef steaks [174] Beef fillets [186] Lamb steaks [187] Beef and pork steaks [188] Bison tenderloin steaks [189]	Vacuum packaging inhibits lipid oxidation, thus preventing unpleasant odors and flavors. Packaged in a modified atmosphere with semi-permeable internal vacuum film leads to an attractive bright red color. Packaging in CO_2_ improves color and the stability of meat color compared to vacuum package.
Active packaging	Pork patties [190] Beef steaks [191] Beef [175,192]	Active packaging does not affect the tenderness of the meat. Desirable bright red color. Incorporation of essential oils in the active packaging leads to unpleasant flavors and aromas.
Edible films and coatings	Beef [176] Pork meat [193] Minced beef [194] Ready to eat (RTE) meat products [178] Pork meat hamburgers [184] Ground-beef patties [195]	The stability of the red color is improved. Lipid oxidation is inhibited, thus preventing unpleasant odor and flavor.
Combination	Pre-cooked convenience-style foods: battered sausages, bacon slices, and meat and potato pies [196]	Optical oxygen sensors in combined vacuum and modified atmosphere packaged and the use of ethanol emitters: ethanol flavor and aroma were not perceived by panelists in two of the three products assessed.

**Table 6 foods-09-01416-t006:** Effects of the addition of some natural ingredients on the sensory characteristics of meat products.

Natural Ingredients		Meat Product	Effects
Essential oils: thyme, oregano, pimento, clove, citron, lemon verbena, lemon, balm, cypress leaf		Lamb meat [203] Raw chicken [204] Meatballs [205] Beef patties [207]	Extracts like clove or cinnamon increase of *L**, *a** and *b** values during storage. Concentrations of essential oils of oregano and thyme greater than 1% led to strong odor and unpleasant taste. Clove extract increased the *a** values.
Plant extracts: grape seed, green tea, pomegranate peel/rind, acerola, pine bark, bearberry, cinnamon bark, rosemary, garlic, oregano, sansho, ginger, sage.	Grape seed extract (GSE) and wine pomace	Low sulfite beef patties [213] Beef enriched with *n*-3 and CLA [139,214] Pork meatballs [215] Chicken nuggets [221]	GSE showed less color changes during the storage. GSE can darken a meat product. No modification of the sensory attributes except for the color.
	Green tea extract	Low sulphite beef patties [213] Pork meatballs [210]	No effects on odor, taste and texture. Degradation of red color is delayed. No modification of the sensory attributes except for the color.
	Rosemary extract	Turkey meat [210] Pork burgers [209]	No effects on sensory characteristics.
	Oregano extract	Sheep burgers [211]	No effects on sensory characteristics.
	Garlic	Low-salt sausages [212]	No effects on aroma, flavor and overall appearance.
	Other fruit extracts: blueberries, raspberry pomace, pomegranate peel and pomegranate juice	Pork burgers and cooked pork ham [216] Beef burgers [217] Chicken patties [3]	Sensory quality was not negatively altered.
	Citrus fiber	Bologne sausage [222]	No effect on color or texture properties. When citrus fiber is combined with rosemary essential oil, the sensory parameters improved.
Spices		Rabbit burgers [206] Indian sheep meat product [208]	Turmeric powder leads to higher yellow values. Cumin and cardamom led to high overall acceptability score.

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
