# Peer review of "The Effects of Processing and Preservation Technologies on Meat Quality: Sensory and Nutritional Aspects"

_foods, 2020, doi:10.3390/foods9101416_

Round 1
Reviewer 1 Report
Article is well written and comprehensive. If possible, i would suggest short additional chapter based on the some new technologies such as electroprocessing, ultrasonics, shock-wave, etc. But it is not necessary as this review paper included all important technologies and some new such as 3D printing.
Author Response
Thank you very much for your helpful comments. The authors have revised the paper accordingly and want to highlight that your comments contribute to improve our manuscript. Please find our response (in green) to reviewer’s specific comments (in black and bold) below.
Article is well written and comprehensive. If possible, i would suggest short additional chapter based on the some new technologies such as electroprocessing, ultrasonics, shock-wave, etc. But it is not necessary as this review paper included all important technologies and some new such as 3D printing.
Response: Thanks for your appropriate suggestions. However, some new technologies such as electroprocessing, ultrasonics or shock-wave have not been included because we wanted to highlight the effects of the most common processing and preservation technologies on meat quality. Despite having included some new technologies such as 3D printing, authors deem that the other new technologies should be addressed in another more specific review.

Reviewer 2 Report
The paper titled 'The effects of processing and preservation technologies on meat quality: sensory and nutritional aspects' is a scientifically sound, comprehensive review on the effect of processing and preservation technologies on meat quality. The review is very well written - well structured, clearly presented, and enriched in tables that summarize the data/ information on the methods described.
I would add only one thing - there is a well organized table presented in the subsection '2.2.5. Reformulation'. Similar tables should be presented for 2.1 and 2.2. sections, to collect the basic data on the methods mentioned, and compare them.
Author Response
Thank you very much for your helpful comments. The authors have revised the paper accordingly and want to highlight that your comments contribute to improve our manuscript. Please find our response (in green) to reviewer’s specific comments (in black and bold) below.
The paper titled “The effects of processing and preservation technologies on meat quality: sensory and nutritional aspects” is a scientifically sound, comprehensive review on the effect of processing and preservation technologies on meat quality. The review is very well written - well structured, clearly presented, and enriched in tables that summarize the data/ information on the methods described.
I would add only one thing - there is a well organized table presented in the subsection '2.2.5. Reformulation'. Similar tables should be presented for 2.1 and 2.2. sections, to collect the basic data on the methods mentioned, and compare them.
Response: Thanks for your appropriate suggestions. Similar tables (new tables 1 and 2) have been presented for 2.1 and 2.2. sections to collect the basic data on the methods mentioned.

Reviewer 3 Report
The present review is very comprehensive and elaborated work on interesting topic.
The abstract should resemble the content of the article more accurately.
Moderate improvement of the style regarding the English language is required. In the Introduction, the reference nr.4 is misquoted.
Diéguez, P.M. et al. (2010) did not work on the obtaining of healthier fresh meat and post-mortem processing as the strategies for the improvement of nutritional properties of new meat products, but on the thermal analysis of meat emulsion cooking process by computer simulation and experimental measurement. Please check if there is more of such mistakes.
The conclusion should be more constructed on the facts presented in the review. Maybe, for an article arranged in such manner, Concluding remarks could be better solution for the final chapter.
Author Response
Thank you very much for your helpful comments. The authors have revised the paper accordingly and want to highlight that your comments contribute to improve our manuscript. Please find our response (in green) to reviewer’s specific comments (in black and bold) below.
The present review is very comprehensive and elaborated work on interesting topic.
Response: Thanks for your comment.
The abstract should resemble the content of the article more accurately.
Response: The abstract content has been rewritten.
Moderate improvement of the style regarding the English language is required.
Response: The English style in the present manuscript have been revised by a native English as recommend by the reviewers.
In the Introduction, the reference nr.4 is misquoted.
Diéguez, P.M. et al. (2010) did not work on the obtaining of healthier fresh meat and post-mortem processing as the strategies for the improvement of nutritional properties of new meat products, but on the thermal analysis of meat emulsion cooking process by computer simulation and experimental measurement. Please check if there is more of such mistakes.
Response: It is reference number 2, which has been incorporated in lines 38 and 171. It mentions the physical changes experienced by meat products and does not talk about fresh products or strategies to obtain healthy products. The authors consider it a mistake.
The conclusion should be more constructed on the facts presented in the review. Maybe, for an article arranged in such manner, concluding remarks could be better solution for the final chapter.
Response: Thanks for your comment. Conclusions has been rewritten.
